# Unveiling China's Forest Soil properties: High-Resolution, Multi-Depth Mapping of Soil Bulk Density and pH Using Machine Learning Methods

5 Jizhen Chen<sup>1,2</sup> Xin Zhang<sup>1,2</sup>, Zihao Fan<sup>1,2</sup>, Tao Liu<sup>3</sup>, Wenfa Xiao<sup>1,2</sup>, Qiwu Sun<sup>4</sup>, Xiangyang Sun<sup>5</sup>, Zilin Huang<sup>1,2</sup>

- <sup>1</sup> Key Laboratory of Forest Ecology and Environment of National Forestry and Grassland Administration, Ecology and Nature Conservation Institute, Chinese Academy of Forestry, Beijing 100091, China
- 10 2 Hubei Zigui Three Gorges Reservoir Forest Ecosystem Observation and Research Station, Zigui 443600, China
  - <sup>3</sup> Department of Earth System Science, Ministry of Education Key Laboratory for Earth System Modeling, Institute for Global Change Studies, Tsinghua University, Beijing 100084, China
  - <sup>4</sup> Research Institute of Forestry, Chinese Academy of Forestry, Beijing 100091, China
  - <sup>5</sup> College of Forestry, Beijing Forestry University, Beijing 100083, China

Correspondence to: Zilin Huang (hzlin66@caf.ac.cn)

**Abstract.** Precise monitoring of key forest soil properties is crucial for addressing global challenges like carbon sequestration and soil acidification. However, existing national soil maps, primarily derived from comprehensive ecosystem samples, inadequately represent the distinct characteristics and high spatial heterogeneity of China's vast and diverse forest ecosystems.

- To bridge this gap, we present the first high-resolution (90-m), forest-specific maps of soil bulk density (BD) and pH across China. Leveraging 4,356 forest soil profiles collected through extensive field surveys and 41 environmental covariates within an optimized Quantile Regression Forests (QRF) framework incorporating forward recursive feature selection (FRFS), we generated wall-to-wall predictions for five standardized depth intervals (0–5, 5–15, 15–30, 30–60, 60–100 cm). Model performance, assessed through 10-fold cross-validation (CV) and independent validation (IV), achieved model efficiency coefficients (MEC) ranging from 0.78 to 0.89 (CV) and 0.60 to 0.66 (IV) for bulk density (BD), and from 0.83 to 0.87 (CV) and 0.71 to 0.81 (IV) for pH, indicating the product's strong capability to capture the spatial variability of forest soil properties across China. The 90-m resolution BD and pH maps contribute to the GlobalSoilMap initiative and provide forest-specific inputs for regional Earth system and land surface models. These products advance the quantification of soil acidification processes and provide critical baseline data for estimating forest soil carbon stocks across China. The dataset is available at
- https://doi.org/10.57760/sciencedb.25375.

#### 1 Introduction

Forest soils are defined as soils that have developed under forest cover, influenced by long-term vegetation—soil interactions, and distinguished by unique physical, chemical, and biological properties (Binkley and Fisher, 2013; Osman, 2013). As key regulators of carbon storage, water cycling, and nutrient availability, forest soils are vital to forest sustainability and policy (Dai et al., 2019; Kleber et al., 2021). China's forest ecosystems span 209 million hectares across diverse climatic zones and complex topographies, encompassing 452 vegetation types to form one of Earth's most ecologically varied forest spectra (Chen et al., 2016; Patton et al., 2019; Zhang et al., 2024). Revealing the spatial distribution of forest soils is fundamental for estimating forest carbon stocks and evaluating forest soil acidification (Zhu et al., 2016; Huang et al., 2022b; Xu et al., 2015). However, forest soils are highly heterogeneous across geographical space, shaped by long-term climatic gradients, vegetation succession, and topographic variation (Zhao et al., 2019; Chen et al., 2022a; Liu et al., 2024). Consequently, accurately revealing the spatial distribution of key forest soil attributes presents a significant challenge.

Digital Soil Mapping (DSM), which integrates machine learning and environmental covariates to predict soil properties across complex landscapes while significantly enhancing spatial soil representation in areas of varied terrain and vegetation, has become a pivotal methodology for acquiring high-resolution spatial soil information (McBratney et al., 2003; Minasny et al., 2013; Padarian et al., 2019). Consequently, numerous countries globally and transnational initiatives have invested substantial resources in using DSM to build national-scale, high-accuracy digital soil databases. These national initiatives typically target resolutions of 90 meters or finer, predicting the spatial distribution of multiple soil attributes across globally standardized depth intervals (0–5 cm, 5–15 cm, 15–30 cm, 30–60 cm, 60–100 cm, and 100–200 cm) as established by GlobalSoilMap.net (Arrouays et al., 2014; Hempel et al., 2014). Exemplary national efforts include the Soil and Landscape Grid of Australia (SLGA; Grundy et al., 2015), France's Soil Data Inventory and Management System (DIGSOL; Mulder et al., 2016), the gSSURGO database in the United States (Ramcharan et al., 2018; Thompson et al., 2020), and the high resolution National Soil Information Grids of China (Liu et al., 2022a; Shi et al., 2025). Concurrently, global-scale initiatives such as SoilGrids provide open-access soil predictions at 250m resolution across all continents using the same GlobalSoilMap standards (Hengl et al., 2017; Poggio et al., 2021). Collectively, these national efforts have substantially advanced our understanding of spatial distribution of multiple soil attributes within their respective coverage areas.

However, a significant limitation persists in characterizing forest soils specifically. Current national-scale soil products in China and globally primarily derive from samples located in comprehensive ecosystem (Poggio et al., 2021; Liu et al., 2022b; Shi et al., 2025). Consequently, they fail to adequately capture the unique physical structures (e.g., higher aggregation, root effects) and biogeochemical processes (e.g., greater susceptibility to acidification driven by vegetation inputs) inherent to forest ecosystems (Widyati et al., 2022; Liu et al., 2024). This creates a critical gap between available soil data products and the urgent need for forest-specific soil information to support accurate carbon stock estimation and acidification risk assessment in these vital ecosystems. Further complicating this gap is the methodological challenge of selecting predictive covariates for forest soil mapping that adequately capture the complex vegetation-soil interactions (Wu et al., 2023).

While the SCORPAN (Soil, Climate, Organisms, Relief, Parent material, Age, and Space) framework underpins digital soil mapping (McBratney et al., 2003; Chen et al., 2022b), optimal covariate selection for forest soils remains challenging due to complex vegetation-soil interactions (Chen et al., 2021; Xue et al., 2025). Model-based feature importance methods, particularly random forest (RF) metrics, have become primary solutions to address dimensionality traps from extensive predictors (Song et al., 2020; Liu et al., 2022a). Subsequent studies have successfully leveraged RF importance to identify key drivers of soil attributes, such as soil-environment relationships via OOB error (Jeune et al., 2018) and critical covariates for soil hydraulic properties (Santos et al., 2023). However, RF-based approaches frequently fail to identify minimal optimal subsets due to variable redundancy. To overcome this, Recursive Feature Elimination (RFE) was developed, which iteratively prunes low-importance features using RF. It distilled high-dimensional covariate sets into parsimonious subsets for soil organic carbon stocks (Hounkpatin et al., 2021) and identified key topographic-vegetation predictors for soil nutrients in heterogeneous (Helfenstein et al., 2024; Shi et al., 2025). Yet RFE's sequential removal risks discarding combinatorially significant variables and incurs high computational costs. More recently, the forward recursive feature selection (FRFS) method has emerged as a superior alternative, excelling at capturing nonlinear relationships while reducing computational costs (Xiao et al., 2022). Xue et al. (2025) successfully applied this method to map the spatial heterogeneity of complex soil attributes across diverse landscapes in China, demonstrating its promising potential for addressing the specific challenges of mapping heterogeneous forest soils. Therefore, our study leverages the FRFS method to tackle the critical covariate selection challenge inherent to mapping China's diverse forest soils.

To address the critical limitations of legacy soil data in representing China's complex and heterogeneous forest ecosystems, we conducted a systematic nationwide forest soil survey. Leveraging machine learning, this study aims to: (1) construct the first nationwide forest-specific soil profile database; (2) develop and apply an optimized DSM framework integrating QRF and FRFS; and (3) pioneer high-resolution (90-meter) digital maps of two fundamental forest soil properties, bulk density (BD) and pH, across China's entire forest domain. Spanning five standardized depth intervals (0–5 cm, 5–15 cm, 15–30 cm, 30–60 cm, and 60–100 cm), these forest-specific maps provide the first continuous, wall-to-wall spatial characterization of BD and pH at 90m resolution, conforming to GlobalSoilMap standards. This unprecedented dataset provides the essential spatial baseline for accurately quantifying forest carbon stocks and assessing soil acidification risks.

#### 90 2 Materials and Methods

We developed 90-m resolution forest soil BD and pH grids for China (0–100 cm) using an optimized QRF model, a machine learning algorithm effective for both spatial prediction and uncertainty quantification (Szatmári et al., 2024). This framework integrated 4,356 georeferenced forest soil profiles, combining historical inventory data (2018–2023). Sampling efforts were designed to ensure ecological and spatial representativeness across major climatic zones and forest types. Soil profiles were harmonized into standardized depth intervals (0–5, 5–15, 15–30, 30–60, and 60–100 cm) using an adaptive equal-area spline method (Bishop et al., 1999; Liu et al., 2022a) and randomly partitioned into training (80%) and independent validation (20%)

subsets. A set of 41 environmental covariates, aligned with soil-forming factors (Jenny, 1941), were resampled to a 90-m grid via bilinear interpolation. Feature selection and hyperparameter tuning were implemented to optimize model performance. Predictive accuracy was evaluated using 10-fold cross-validation and independent validation based on a withheld dataset. A summary of the modelling framework is shown in Figure 1.

Figure 1. Workflow diagram for forest soil mapping.

## 2.1 Data compilation

#### 2.1.1 Soil database

We developed a comprehensive forest soil property database for China, representing the most extensive and up-to-date collection of forest soil data to date. The data were compiled from two major nationwide forest soil surveys conducted in 2018 and 2023, complemented by independently conducted regional forest soil surveys during the intervening years to enhance spatial and ecological representation. These surveys employed a stratified sampling design to ensure broad representativeness across China's major forest ecosystems, covering diverse climate zones, forest types, and topographic gradients. In addition to these national efforts, data from independently conducted regional forest soil surveys during the intervening years were also incorporated to enhance spatial and ecological representation. After rigorous quality control and data harmonization, the final integrated dataset comprises 8,709 soil profiles and 18,193 soil samples. Of these, 4,356 profiles and 11,873 samples contain both BD and pH values, forming the core dataset used in this study. The spatial distribution of sampling plots and forest coverage is displayed in Figure 2.

To ensure data comparability and minimize measurement errors, all samples were processed under identical conditions. Soil sampling and analysis followed standardized protocols to ensure data consistency. Soil samples were collected using a soil auger, air-dried at room temperature, homogenized, and passed through a 2 mm sieve to remove coarse fragments and roots for physicochemical analyses. Undisturbed soil cores were collected from each horizon using a cutting ring sampler to determine BD. Soil pH was measured using a pH meter following the potentiometric method, with a soil-to-water ratio of 1:2.5 (w/v). Reference materials were used throughout the analytical process to ensure measurement accuracy and control data quality.

Figure 2. Spatial distribution of soil sampling plots and forest coverage. Publisher's remark: please note that the above figure contains disputed territories.

## 2.1.2 Standard soil depths

125

Following GlobalSoilMap specifications (Arrouays et al., 2014), soil samples are typically standardized to fixed depth intervals of 0–5, 5–15, 15–30, 30–60, 60–100, and 100–200 cm. To model continuous depth functions from soil property measurements recorded by genetic horizons, equal-area quadratic spline interpolation is commonly used (Bishop et al., 1999). However, natural soil profiles often contain abrupt changes in properties between adjacent horizons, leading to inconsistencies with these standardized depth layers. To address this issue and reduce fitting errors, we applied an adaptive equal-area spline method (Liu et al., 2022a). This method detects abrupt transitions by calculating the ratio of property values between adjacent horizons and applying a predefined threshold. When such discontinuities are identified, a 1 cm transitional layer is inserted between the affected horizons before spline fitting. This adjustment ensures improved consistency with the observed morphological

structure of each soil profile. While the GlobalSoilMap framework includes the 100–200 cm interval, our study focused on the upper five layers (0–5, 5–15, 15–30, 30–60, and 60–100 cm) due to the limited number of forest soil profiles extending beyond 100 cm in depth.

#### 2.1.3 Environmental Covariates

Soil formation is governed by the combined effects of climate, parent material, topography, vegetation, and human activities.

In this study, 41 environmental covariates were selected based on the soil-forming factor framework (Jenny, 1941; Minasny et al., 2013) and categorized into five groups: parent material, climate, organisms, topography, and intrinsic soil properties (Table S1). To reduce multicollinearity, a variance inflation factor (VIF) threshold of less than 10 was applied through iterative variable exclusion.

All covariate layers were projected using the Albers Equal Area coordinate system (EPSG:4326, WGS84 datum) and resampled to a unified 90-m spatial resolution via bilinear interpolation. For multi-year variables, long-term annual means and growing season (May to September) averages were calculated from monthly records spanning 2003 to 2023, thereby capturing both historical trends and contemporary environmental conditions relevant to forest soil development.

Climate-related covariates included temperature, precipitation, potential evapotranspiration, and solar radiation, derived from the National Tibetan Plateau Data Center (https://data.tpdc.ac.cn) and the TerraClimate dataset. Parent material characteristics were obtained from Sentinel-2 imagery using the shortwave infrared band (B12) and the B8/B12 band ratio to estimate clay content. Depth to Bedrock (DTB) data were incorporated to represent weathering intensity, and lithological context was supplemented using the Geological Map of China. Topographic attributes were extracted from the NASADEM digital elevation model (https://lpdaac.usgs.gov/products/nasadem\_hgtv001/) and computed using SAGA GIS (http://www.saga-gis.org). Vegetation indicators were sourced from MODIS products, including NDVI, NDWI, LAI, and NPP, while forest type classifications were based on the National Atlas of Forest Vegetation in China.

## 2.2 Modelling

## 2.2.1 Covariate selection

To balance model parsimony with biogeochemical interpretability, we adapted the Forward Recursive Feature Selection (FRFS) approach proposed by Xiao et al. (2022) into a depth-specific selection framework, applied independently to four standardized soil layers. The procedure comprised three sequential steps. First, the covariate with the highest predictive importance, as assessed by permutation-based Random Forest analysis, was selected to initiate the model. Subsequently, additional variables were iteratively added based on two criteria: a reduction of more than 5% in five-fold cross-validated root mean square error (RMSE) and a variance inflation factor (VIF) below 10. The selection process was automatically terminated when five consecutive iterations failed to achieve an RMSE improvement of at least 1%, thereby avoiding model overfitting. This

180

185

hierarchical strategy ensured effective dimensionality reduction while maintaining predictive performance across all soil depths. The framework was applied across four distinct soil horizons.

#### 2.2.2 Predictive models

Quantile Regression Forests (QRF), a nonparametric ensemble learning method extending the Random Forest framework, were used to model the relationships between environmental covariates and soil properties, while explicitly quantifying predictive uncertainty (Meinshausen, 2006). As a state-of-the-art algorithm in DSM (Liu et al., 2022a; Poggio et al., 2021; Pouladi et al., 2019), QRF leverages both bootstrap aggregation of regression trees and randomized feature subset selection at each node, enabling robust handling of high-dimensional, non-stationary data.

Unlike standard Random Forests, QRF retains the full conditional distribution  $F(y \parallel X = x)$  At each prediction node, allowing estimation of both point predictions and confidence intervals. This is achieved via kernel-based empirical distribution construction:

$$\hat{F}(y|X=x) = \sum_{i=1}^{n} w_i(x) I(y_i \le y) \tag{1}$$

where  $w_i(x)$  Is the weight assigned to each training observation based on terminal node proximity. The conditional quantile function is derived as:

$$\widehat{q}_{\alpha}(x) = \inf\{y : \widehat{F}(y \mid X = x) \ge \alpha\} \tag{2}$$

for a given quantile level  $\alpha \in (0,1)$ . This allows the derivation of the median estimate  $\hat{q}_{0.5}(x)$ , prediction intervals  $\left[\hat{q}_{a/2}(x), \hat{q}_{1-a/2}(x)\right]$ , and the full uncertainty distribution, enhancing both interpretability and decision support in forest soil assessments.

where specifies the quantile level (e.g.,  $\alpha=0.5$  or median prediction). This formulation generates three interconnected outputs: the median predictor as a robust central tendency estimate, prediction intervals for heteroscedastic uncertainty quantification, and the complete conditional distribution through parametric evaluation of  $\hat{q}_{\alpha}(x)$  across the  $\alpha$  continuum.

To implement QRF across China's forested regions, we adopted a spatially parallel computing framework. The study area was divided into 461 contiguous grid tiles ( $10 \times 10 \text{ km}$ ) using the Albers Equal Area projection. Model execution was carried out using the quantregForest package in R 4.2.1, running on 24 logical cores of a high-performance computing node. Spatial continuity was preserved across grid boundaries using a Gaussian kernel-based edge matching algorithm, enabling seamless 90-m resolution prediction without artifacts.

#### 2.2.3 Hyperparameter tuning

Hyperparameter optimization was conducted for three parameters critical to model performance: mtry (number of variables randomly sampled at each split), num.trees (number of trees), and min.node.size (minimum samples per terminal node). The randomized search strategy was employed, guided by 10-fold cross-validation and using RMSE as the evaluation metric. Fifty

iterations of parameter space sampling were performed to identify the optimal combination. Final hyperparameter values were selected based on configurations that yielded the highest prediction accuracy on the validation dataset. A summary of optimized parameters for each soil property and depth interval is provided in Table S2.

#### 2.3 Model validation

To comprehensively evaluate model performance, we applied two complementary validation strategies: 10-fold crossvalidation on the training dataset (80%) and independent validation using a held-out test set (20%). These schemes were implemented across the entire study region to assess the predictive accuracy of forest soil BD and pH.

In 10-fold cross-validation, the training set was randomly partitioned into ten equal subsets. In each iteration, nine subsets were used to train the model, and the remaining one was used for validation. This procedure was repeated ten times, ensuring each subset served as validation data exactly once. Model accuracy was assessed by averaging performance metrics across folds, including mean error (ME), root mean square error (RMSE), and the model efficiency coefficient (MEC).

For independent validation, the reserved test set was excluded entirely from model training and hyperparameter tuning, thereby providing an unbiased evaluation of generalizability. The formulas used for calculating the evaluation metrics are as follows:

$$ME = \frac{1}{n} \sum_{i=1}^{n} (\hat{y}_i - y_i)$$
(3)

210 RMSE = 
$$\sqrt{\frac{1}{n}\sum_{i=1}^{n}(y_i - \hat{y}_i)^2}$$
 (4)

$$MEC = 1 - \frac{\sum_{i=1}^{n} (y_i - \hat{y}_i)^2}{\sum_{i=1}^{n} (y_i - \hat{y})^2}$$
 (5)

where  $y_i$  is the observed soil property value,  $\hat{y_i}$  is the predicted value, and  $\bar{y}$  is the mean of observed values. ME, also referred to as bias, measures average deviation. RMSE reflects the overall prediction error, with lower values indicating higher accuracy. MEC, equivalent to the coefficient of determination (R<sup>2</sup>), ranges from 0 to 1, with higher values indicating better predictive performance.

#### 2.4 Uncertainty Quantification

Quantifying spatial uncertainty is essential in DSM, as prediction errors may arise from input data variability, model structure, and environmental heterogeneity (Arrouays et al., 2014; Poggio et al., 2021; Liu et al., 2022a; Shi et al., 2025). To visualize the spatial distribution of prediction uncertainty, we calculated the Prediction Interval Ratio (PIR), defined as the ratio between the 90% prediction interval width and the median estimate:

$$PIR = \frac{q_{0.95} - q_{0.05}}{q_{0.50}} \tag{6}$$

where  $q_{0.95}$  and  $q_{0.05}$  represent the upper and lower bounds of the 90% prediction interval, respectively, and  $q_{0.05}$  denotes the median prediction. PIR is a dimensionless metric that quantifies the relative spread of prediction uncertainty around the central estimate. Higher PIR values indicate greater dispersion and, therefore, higher predictive uncertainty.

To evaluate the calibration of these uncertainty estimates, we used the Prediction Interval Coverage Probability (PICP), computed from the independent validation dataset (Goovaerts, 2001). PICP measures the proportion of observed values that fall within their corresponding prediction intervals at a specified confidence level (e.g., 90%). A well-calibrated model should yield a PICP value close to the nominal coverage. For example, a 90% prediction interval is considered reliable if the empirical PICP also approximates 90%. Systematic deviation from this benchmark can indicate miscalibration: a PICP above the target level suggests that intervals are too narrow (underestimated uncertainty), while a PICP below the target indicates overly wide intervals (overestimated uncertainty) (Poggio et al., 2021; Liang et al., 2019). This diagnostic approach supports the robust interpretation of uncertainty in DSM outputs.

#### 3 Results

#### 3.1 Forest soil database overview

Table 1 presents the harmonized forest soil database comprised 4,356 forest soil profiles distributed across China. Using the equal-area spline method, soil property values were standardized to fixed depth intervals (0–5, 5–15, 15–30, 30–60, and 60–100 cm), resulting in 15,845 horizons for BD and 15,978 horizons for pH. BD showed low skewness across depths (0.16–0.42), while pH closely followed a normal distribution (skewness 0.05–0.19). Mean values of both BD and pH increased gradually with depth, from 1.206 to 1.342 g/cm³ for BD and from 6.07 to 6.47 for pH. The standard deviation of BD increased gradually from 0.261 in the shallowest layer (0–5 cm) to 0.308 in the lowest depth interval (60–100 cm), while pH showed a more pronounced rise in variability, with its standard deviation increasing from 0.909 to 1.327 across the same range. Both parameters showed wide value ranges across all depths (BD: 0.15–2.30 g/cm³; pH: 4.00–8.70).

Table 1. Statistical summary of BD and pH at five depth intervals. Refer to Table 1 for the abbreviations and units of the soil properties interested.

| Property | Depth (cm) | $N^{a)}$ | Mean  | SD    | Min   | Max   | Skewness |
|----------|------------|----------|-------|-------|-------|-------|----------|
| BD       | 0–5        | 4356     | 1.206 | 0.261 | 0.152 | 2.057 | 0.162    |
|          | 5–15       | 3522     | 1.209 | 0.288 | 0.284 | 2.291 | 0.317    |
|          | 15–30      | 3488     | 1.287 | 0.269 | 0.301 | 2.271 | 0.422    |
|          | 30–60      | 2973     | 1.340 | 0.269 | 0.257 | 2.215 | 0.393    |
|          | 60–100     | 1506     | 1.342 | 0.308 | 0.534 | 2.291 | 0.367    |
| рН       | 0–5        | 3963     | 6.066 | 0.909 | 4.000 | 8.440 | 0.045    |
|          | 5–15       | 3962     | 6.131 | 0.991 | 4.000 | 8.515 | 0.111    |

|   | 15–30  | 3816 | 6.172 | 1.005 | 4.030 | 8.420 | 0.087 |
|---|--------|------|-------|-------|-------|-------|-------|
|   | 30-60  | 2783 | 6.458 | 1.198 | 4.040 | 8.640 | 0.194 |
| ( | 60–100 | 1454 | 6.466 | 1.327 | 4.040 | 8.704 | 0.119 |

a) N varies slightly between properties and depths due to sample availability for specific analyses.

#### 3.2 Model performance

The performance of the QRF models was evaluated after training and optimisation. Table 2 lists the 10-fold cross-validation (CV) and independent validation (IV) results for BD and pH predictions of our study across five soil depth intervals. Model performance varied with specific soil properties. For BD, 10-fold CV achieved high accuracy (MEC = 0.782-0.889, RMSE = 250 0.079-0.090 g/cm³), explaining 78.2-88.9% of variability. IV yielded robust but reduced performance (MEC = 0.598-0.657, RMSE = 0.155-0.181 g/cm<sup>3</sup>), retaining 59.8-65.7% explanatory power. Conversely, pH predictions demonstrated superior accuracy: CV maintained strong performance across depths (MEC = 0.834-0.868, RMSE = 0.214-0.254), peaking at 60-100 cm (MEC = 0.868, RMSE = 0.238; 86.8% variability explained). IV confirmed generalizability (MEC = 0.705-0.812, RMSE = 0.432 - 0.515).

Model performance also varied with depth. BD prediction accuracy exhibited non-monotonic depth dependence during independent validation, peaking at intermediate depths (15-30 cm: MEC = 0.657) with lower accuracy in surface layers (0-5 cm: MEC = 0.598) and deep layers (60-100 cm: MEC = 0.656). In contrast, pH prediction accuracy systematically increased with soil depth under both validation frameworks. CV showed MEC progression from 0.844 (0-5 cm) to 0.868 (60-100 cm), while IV demonstrated improvement from 0.705 (0-5 cm) to 0.812 (60-100 cm). Optimal pH performance consistently 260 occurred at the deepest interval (60-100 cm) for both validation methods. In addition, all predictions maintained negligible bias ( $|ME| \le 0.019$ ) across depth intervals.

Table 2. Predictive performance of BD and pH predictions.

| Validation | Depth (cm) | 10-fold CV |       |        | IV    |       |        |
|------------|------------|------------|-------|--------|-------|-------|--------|
|            |            | MEC        | RMSE  | ME     | MEC   | RMSE  | ME     |
|            | 0–5        | 0.782      | 0.090 | 0.000  | 0.598 | 0.164 | -0.01  |
|            | 5-15       | 0.815      | 0.084 | 0.000  | 0.611 | 0.181 | -0.017 |
| BD         | 15-30      | 0.828      | 0.081 | -0.000 | 0.657 | 0.155 | 0.006  |
|            | 30-60      | 0.874      | 0.079 | -0.000 | 0.614 | 0.166 | 0.005  |
|            | 60-100     | 0.889      | 0.087 | 0.000  | 0.656 | 0.166 | -0.019 |
| рН         | 0–5        | 0.844      | 0.215 | 0.000  | 0.705 | 0.432 | -0.003 |
|            | 5-15       | 0.834      | 0.254 | 0.000  | 0.726 | 0.480 | -0.001 |
|            | 15-30      | 0.854      | 0.214 | 0.000  | 0.742 | 0.448 | -0.007 |
|            | 30-60      | 0.854      | 0.256 | 0.001  | 0.760 | 0.515 | -0.002 |
|            | 60-100     | 0.868      | 0.238 | 0.001  | 0.812 | 0.492 | 0.014  |

## 3.3 Spatial patterns

#### 3.3.1 Spatial patterns of BD

The BD maps predicted by QRF (Figure 3) show consistent mean values ranging from 1.16 to 1.34 g/cm³ and standard deviations of 0.15–0.21 g/cm³ across all soil depths (Tables S3 and Fig S3). Macroscale patterns align with CSDLv2, ChinaSoilInfoGrids, and SoilGrids 2.0 (Fig. S1).

Spatially, BD exhibits distinct regional variation. The highest values occur in southwestern China (mean BD = 1.45 g/cm³) and the lowest values in northeastern China (mean BD = 0.79 g/cm³). Southwestern China consistently forms the highest-value area across all soil layers. Northeastern China constitutes the lowest-value zone. In eastern China, BD values increase from the coast inland. Across the eastern coastal and southern regions, BD gradients occur from south to north and from coast to inland.

Vertically, BD increases with depth. Surface layers (0–5 cm) show the lowest BD, with minimal values concentrated in northeastern and southeastern coastal regions (mean values 0.79 g/cm³ and 1.19 g/cm³, respectively). High-BD zones in the southwest expand slightly with depth. Within the middle soil depths (5–15 cm and 15–30 cm), spatial variability intensifies: low-BD zones extend from the northeast into North China, alongside distinct high-BD cores in the southwest. The 30–60 cm layer reaches the highest mean BD (1.32 g/cm³). The deep soil layer (60–100 cm) has a mean BD of 1.23 g/cm³, featuring extensive high-BD areas in southwest China and reduced low-BD coverage in northeastern areas.

Figure 3. The predicted maps of predicted BD at 0–5 cm (a), 5–15 cm (b), 15–30 cm (c), 30–60 cm (d) and 60–100cm (e) depths.

Publisher's remark: please note that the above figure contains disputed territories.

https://doi.org/10.5194/essd-2025-496 Preprint. Discussion started: 5 November 2025 © Author(s) 2025. CC BY 4.0 License.

## 3.3.2 Spatial patterns of pH

The predicted pH maps based on QRF are illustrated in Figure 4.. These maps showed mean values ranging from 5.70 to 6.06 and standard deviations ranging from 0.65 to 0.81 across all depths (Tables S3 and Fig.S4). Macroscale patterns align with those of CSDLv2, ChinaSoilInfoGrids, and SoilGrids 2.0 (Fig. S2).

Spatially, pH shows regional differentiation. Forest soils in South and Southwest China exhibit lower pH values (pH  $\leq$  5.76). Forest soils in North and Northwest China exhibit higher pH values (pH  $\geq$  6.50). Northeast China shows intermediate pH values, ranging between those of the southern and northern regions.

Vertically, pH patterns show both consistency with surface layers and changes with depth. Surface layers (0–5 cm, 5–15 cm) in the South and Southwest show the lowest pH values. With increasing soil depth (15–30 cm, 30–60 cm, and 60–100 cm), pH values in the southern regions increase. pH values in the northern regions become more stable with depth, showing reduced range. Overall, spatial variability in pH decreases in deeper soil layers.

Figure 4. The predicted maps of predicted pH at 0–5 cm (a), 5–15 cm (b), 15–30 cm (c), 30–60 cm (d) and 60–100cm (e) depths. Publisher's remark: please note that the above figure contains disputed territories.

https://doi.org/10.5194/essd-2025-496 Preprint. Discussion started: 5 November 2025 © Author(s) 2025. CC BY 4.0 License.

#### 3.4 Prediction uncertainty

Visualization of prediction uncertainty using the PIR highlighted clear regional variations in uncertainty for BD and pH across China. Higher uncertainty for BD was concentrated in the northeastern and southwestern regions, while lower uncertainty characterized the southeastern coastal areas (Fig. S5). Conversely, pH uncertainty was more pronounced in northern China and parts of the southwest, with relatively lower levels observed in the northeast and the central-eastern coastal zone (Fig. S6). Overall, areas of elevated uncertainty predominantly coincided with southwestern China, where complex soil-landscape interactions likely contribute to increased model uncertainty. Additionally, regions with sparse data coverage, such as high-altitude areas, exhibited amplified extrapolation uncertainty due to limited representation in the training dataset, further challenging model reliability in these environments. For both BD and pH, prediction uncertainty generally increased with soil depth, a pattern potentially attributable to the reduced availability of soil observations at deeper intervals.

To ensure that biased uncertainty estimates do not compromise practical applications of the model, we further employed the PICP to perform this critical validation step. Five predictive accuracy plots were generated to evaluate the alignment of predicted intervals with actual observations for BD and pH (Figure 5). The QRF-based digital soil mapping model showed close adherence to the 1:1 reference line across both properties, indicating strong consistency in local uncertainty estimation. However, for pH, a slight overestimation of uncertainty was detected at intermediate probability levels (60%–90%) within subsurface layers (0–60 cm), suggesting minor deviations from optimal calibration. In contrast, uncertainty quantification for BD remained well-calibrated across all depth intervals and probability thresholds.

Figure 5. Validation of uncertainty quantifications.

## 315 **3.5 Covariate importance**

Relative importance of the environmental covariates used in the soil spatial predictions is shown in Figure 6. The FRFS framework enabled depth-specific dimensionality reduction, retaining 7 to 16 covariates per soil layer while eliminating 60.98% to 77.93% of the initial set (Table S4). Considerable variation in covariate relative importance was observed both between soil properties and across depths.

For forest soil BD prediction in the ensemble model, PRE\_A showed the highest relative importance across all depths (0-100 cm), ranging from 11.41% to 23.73%. Other predictors exhibited clear depth-dependent variations in relative importance. In surface soils (0–15 cm), NDWI\_A, NPP, Elevation, and Soil had notable relative importance. In middle layers (15–60 cm), the relative importance of NPP and Soil increased. In deep layers (60-100 cm), PRE\_A, NDWI\_A, and NPP maintained high relative importance, while parent material (PM) and bedrock depth (DTB) showed increased relative importance. Analysis by

factor category (Fig. S7) indicated that climate factors contributed the most to BD prediction across all depths (44.00%–62.47%), significantly exceeding the contributions from other factor categories.

For soil pH prediction in the ensemble model, the synergistic combination of NDWI\_A, PRE\_A, and NPP showed the highest relative importance across the entire soil profile (29.11–36.14%). Secondary factors exhibited depth-dependent variations in their relative importance. In surface soils (0–15 cm), vegetation indicators (LAI\_A, NDVI\_MAX) had the strongest secondary relative importance. In middle layers (15–60 cm), the relative importance of topographic factors (Elevation, Geomor) and parent material (PM) increased. In the deep soil layer (60–100 cm), NDWI\_GS and ALR2 showed increased relative importance, while DTB maintained consistently high relative importance. Analysis by factor category (Fig. S7) indicated that organism factors (primarily vegetation-related) contributed the most to the prediction (up to 36.14%), followed by varying relative contributions from climate and parent material factors across depths.

Figure 6. Variable importance for model training at different soil depths. The abbreviations of the predictors are defined in Table S1.

365

#### 4 Discussion

#### 4.1 Model performance improvement

China has established valuable national soil datasets, such as the comprehensive CSDLv2 (Shi et al., 2025) and the highresolution ChinaSoilInfoGrids (Liu et al., 2022a), which provide robust insights into soil properties across diverse ecosystems (Table S5). Building upon these foundations and focusing specifically on forest ecosystems, our work developed a model to simulate the spatial distribution of BD and pH within China's forests. As reported in Sect.3.2, the forest-optimized model demonstrated reliable performance for this targeted application under both CV and IV frameworks. Critically, the IV results, based solely on external forest samples withheld from model development, confirmed the model's robustness and generalizability within the specific context of China's forest ecosystems. These outcomes underscore the effectiveness of our approach in capturing forest-specific soil patterns.

Our methodology integrates two key innovations specifically designed to address the unique challenges of DSM in forest ecosystems. First, recognizing that forest soil sites are often underrepresented in large-scale national databases (which typically prioritize agricultural land) (Liu et al., 2022a; Shi et al., 2025), we constructed China's first dedicated, systematic forest soil dataset. This dataset, encompassing dominant forest types across major ecoregions, provides the essential foundation for characterizing forest-specific paedogenic processes and deriving accurate soil-environment relationships reflecting forest biogeochemistry.

Second, forest covariate optimization contributed significantly to the improved accuracy (Table S5). Recent DSM studies 355 employing the SCORPAN framework have highlighted that relying solely on universal predictors may overlook critical ecosystem-specific variations, particularly in heterogeneous regions (Sun et al., 2022; Zhang et al., 2025). Building on this insight, our study explicitly incorporates forest-specific drivers to enhance mapping accuracy across China's complex landscapes. By tailoring covariate selection to forest ecosystems, we overcome the cross-ecosystem extrapolation bias prevalent in current national datasets. For instance, forest soil BD in China is mainly influenced by root-mediated aggregation (Liu et al., 2019; Zheng et al., 2023), while agricultural BD reflects region-specific tillage practices, such as the puddling effects in rice-wheat rotations (Hou et al., 2012). Similarly, forest soil pH is controlled by litterfall chemistry with distinct stoichiometry in Chinese subtropical forests (Zhou et al., 2016; Farooq et al., 2022), whereas agricultural pH is dominated by long-term nitrogen fertilization (Wang et al., 2019; Jia et al., 2022). Consequently, moving beyond universal predictors to select covariates specific to forest ecosystems was fundamental to the improved mapping accuracy.

Consequently, our forest-optimized framework, underpinned by the purpose-built dataset and ecologically informed covariates, generates a specialized and complementary perspective on soil property mapping for forest ecosystems. It delivers China's first comprehensive, 90-m resolution wall-to-wall maps characterizing forest BD and pH spatial patterns. These results validate the critical importance of incorporating ecosystem context into digital soil mapping (Padarian et al., 2019).

### 4.2 Potential applications

The high-resolution spatial dataset of forest soil BD and pH developed in this study represents a national-scale digital soil mapping product that captures the spatial variability of key soil physical and chemical properties across forested regions in China. Accurate knowledge of BD and pH is fundamental for estimating soil carbon stocks (Batjes, 2016), and monitoring forest ecosystem responses to land-use and climate change (Pan et al., 2011). Bulk density is critical for accurate estimation of soil carbon stocks, while pH governs nutrient availability, microbial activity, and forest productivity (Liu et al., 2024), and is significant for understanding forest soil acidification (Farooq et al., 2022). The dataset fills longstanding gaps in forest soil data coverage in China, and supports applications in ecosystem assessment and long-term soil monitoring. Beyond its scientific value, this product contributes to national strategies on carbon neutrality and ecological restoration and aligns with international environmental commitments including the UN Decade on Ecosystem Restoration and the Sustainable Development Goals (UNEP, 2021; IPCC, 2022).

#### 380 4.3 Limitations and outlook

Our study advances high-resolution DSM in forest ecosystems, yet several methodological limitations remain and merit further investigation, particularly regarding the predictive reliability of machine learning approaches. Machine learning, while significantly enhancing DSM through capturing nonlinear soil-environment relationships, are constrained by limitations in spatial coverage and feature-space representativeness (Yang et al., 2013; Chen et al., 2019). Forests exhibit pronounced landscape heterogeneity, complicating sampling design and frequently resulting in imbalanced training datasets (Huang et al., 2022a; Liu et al., 2022b; Shao et al., 2022). As demonstrated by Westhuizen et al. (2024), models trained on such datasets yield biased predictions in undersampled regions. Although ensemble methods manage uncertainty in sparse data settings, they may prioritize statistical regularities over mechanistic soil formation processes (Sylvain et al., 2021; Liu et al., 2022b). Emerging hybrid frameworks integrating environmental similarity metrics with pedological expertise show promise in addressing these challenges (Zhao et al., 2024), though their scalability requires further validation (Miranda et al., 2023; Potash et al., 2023; Rodrigues et al., 2025). Specifically, strategic sampling designs incorporating stratified and adaptive approaches across diverse forest landscapes and soil types are crucial to mitigate dataset imbalance and capture underlying heterogeneity (Brus et al., 2011). Concurrently, exploring novel covariates derived from multi-source remote sensing (e.g., hyperspectral, LiDAR, radar) and proximal sensing (Xue et al., 2025), alongside improved representations of depth-dependent properties and long-term environmental legacies, could substantially enrich the feature space and better characterize the complex soil-forming factors operating in forest ecosystems (Vaysse and Lagacherie, 2017; Wadoux et al., 2020). Integrating such refined datasets within hybrid modeling frameworks holds considerable potential for improving the accuracy and reliability of forest DSM predictions.

#### 5 Data and code availability

The soil property maps generated in this study include soil pH and BD for five depth intervals (0–5 cm, 5–15 cm, 15–30 cm, All resources for the ensemble machine learning model, including training and testing code, are publicly available at https://github.com/cjz-ux/China\_forest\_DSM/tree/main. The soil property maps generated in this study include soil pH and BD for five depth intervals (0–5 cm, 5–15 cm, 15–30 cm, 30–60 cm, and 60–100 cm), with a spatial resolution of 90 meters. These maps are openly accessible via the platform link: https://doi.org/10.57760/sciencedb.25375 (last access: 19 September 2025) (Chen et al., 2025). Users can download the datasets efficiently using the provided FTP credentials and any standard FTP client.

#### 6. Usage note

It is important to highlight that uncertainties associated with the spatial predictions of soil pH and BD have been not only quantified but also explicitly embedded in the corresponding maps. These uncertainty estimates offer critical insights into the reliability of predictions. Users are strongly encouraged to interpret the pH and BD maps alongside their respective uncertainty layers to ensure scientific rigor in downstream analyses and to support evidence-based decision-making and policy formulation. The inclusion of uncertainty information should not be regarded as a drawback. In fact, the adoption of standardized protocols for uncertainty quantification and reporting, which are now commonly used in DSM, enhances the transparency and applicability of the dataset. Users should also be aware that no spatial map represents a perfect depiction of reality. Interpreting these predictions without considering uncertainty introduces scientific and practical risks. The uncertainty layers serve as a guide for context-sensitive interpretation.

The current version of the China forest soil pH and BD grids is based on soil sampling limited to mainland China. Data from Hong Kong, Macau, and Taiwan are not included due to availability constraints. While this spatial extent reflects the existing sampling framework, future updates will aim to incorporate broader geographic coverage. Users should clearly acknowledge this limitation when applying the dataset in regional-scale modelling or policy-oriented analyses.

In addition, the environmental covariates used in the DSM workflow exhibit spatially heterogeneous coverage, with localized data gaps in certain regions (e.g., areas with steep elevation gradients or low-quality remote sensing input). To ensure model reliability, soil property predictions were restricted to areas where all covariates are fully available. Consequently, regions with missing covariate data were excluded from the final maps. Users should check the alignment of their study area with the covariate intersection mask, which is provided as ancillary metadata, to confirm the spatial applicability of the dataset.

#### 7 Conclusions

Our study developed the first high-resolution mapping of forest soil BD and pH across China, leveraging forest soil profiles from the latest national forest soil survey. We achieved this detailed characterization across complex forest soil landscapes by

https://doi.org/10.5194/essd-2025-496 Preprint. Discussion started: 5 November 2025 © Author(s) 2025. CC BY 4.0 License.

integrating the predictive soil mapping paradigm with FRFS, QRF, and a detailed suite of forest-specific soil-forming environmental factors within a high-performance parallel computing environment. This integrated approach not only effectively reduced errors and training time but also enhanced the performance of the final predictive models. The resultant multilayer maps delineate pronounced regional gradients and fine-scale forest soil heterogeneity across depths, outperforming existing products in accuracy, spatial detail, and provision of local uncertainty metrics. These high-resolution forest soil property maps represent a contribution to the GlobalSoilMap.net project and provide critical baseline data for China's forest carbon accounting and understanding of soil acidification processes.

## Author contributions.

Conceptualization: JZC; Data curation: QWS, XYS, JZC, ZHF, XZ, and ZLH; Formal analysis: JZC; Funding acquisition: ZLH and WFX; Methodology: JZC and XZ; Supervision: JZC, ZLH, and WFX; Validation: JZC; Writing – original draft preparation: JZC, ZLH, and TL; Writing – review & editing: JZC, ZLH, and TL.

#### 440 Competing interests.

The contact author has declared that none of the authors has any competing interests.

#### Acknowledgements.

This work was supported by the National Key Research and Development Program of China (No.2021FY100800).

We would like to express our gratitude to Professor Feng Liu at the Institute of Soil Science, Chinese Academy of Sciences

(Nanjing, China), for his valuable suggestions that contributed to this study.

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
