# Peer review of "Unveiling China's Forest Soil properties: High-Resolution, Multi-Depth Mapping of Soil Bulk Density and pH Using Machine Learning Methods"

_Earth System Science Data, 2025_

## Referee Comment (RC2)

The authors address a significant topic by mapping key soil properties across China's forests using a comprehensive dataset. The resulting high-resolution products have the potential to be a valuable resource for the scientific community. However, several methodological and descriptive aspects require substantial improvement to ensure the reliability and reproducibility of the findings. I recommend a major revision to help the manuscript reach the high standards required for publication. My main comments on the manuscript are as follows:

General comments:

1. Resampling covariates with diverse native resolutions to a 90-m grid introduces significant uncertainty, particularly for inputs derived from coarser scales. This issue warrants a detailed discussion and quantification to assess the reliability of the final high-resolution maps.

2. The manuscript provides insufficient discussion on the variable importance for BD and pH across different soil layers. The underlying reasons for these variations require further elaboration.

3. Natural and planted forests possess distinct driving mechanisms. Developing separate models for each forest type is advisable to accurately capture these specific variations.

4. Providing spatial distribution maps for every covariate listed in Table S1 is advisable. The figures need to clearly display the value ranges for continuous variables and the distinct spatial patterns for each category within categorical variables. Specifying the number of sample points in both the training and validation sets for each categorical variable is recommended.

5. Forest age represents a critical covariate. Incorporating this variable into the analysis is advisable to improve model performance.

6. Providing the original data is necessary to facilitate the reproducibility of the study by other researchers.

7. Comparing the current results with existing soil BD and pH products is recommended. The manuscript needs to clarify specific improvements and explain the reasons for these advancements.

8. Presenting the spatial distribution of sample points for both the training and validation sets is necessary. The manuscript should also address whether these distributions are spatially balanced.

9. A more detailed description of the raw data is necessary. The manuscript should specify the sample sizes and spatial distributions across different temporal periods, soil types, and forest types.

Minor comments:

1. Lines 1-3: The general phrase "Soil properties" creates redundancy with the specific variables "Bulk Density and pH," necessitating a more concise revision such as "High-Resolution, Multi-Depth Mapping of Soil Bulk Density and pH in China's Forests Using Machine Learning"

2. Lines 20-21: The claim of being 'first' is inaccurate due to the existence of prior 90-m products, so the text should be revised to focus on the specific contribution to forest ecosystems instead.

3. Lines 73-74: The phrase "in heterogeneous" is grammatically incorrect.

4. Lines 109-111: This sentence is redundant and should be deleted.

5. Lines 111-112: Specify the quality control and data harmonization methods.

6. Lines 139-155: The 41 environmental covariates lack necessary citations, and the sources or the data itself should be made accessible to readers to ensure reproducibility.

7. Lines 159-160: The number of standardized soil layers should be corrected from four to five.

8. Lines 222-223: $q_{0.50}$ denotes the median prediction.

9. Lines 429-430: The use of "first" is an absolute claim that is prone to dispute.

---

## Author Comment (AC1)

**RESPONSE TO REVIEWER #1**

The manuscript proposes a high-resolution forest-specific mapping approach for predicting soil bulk density and pH across China. It presents a substantial body of work and addresses a topic of interest, which has the potential to contribute to the field. However, in my opinion, the current manuscript requires major revision before it can be considered for publication.

Overall, the manuscript is informative and holds value but requires further refinement. The authors are encouraged to more clearly emphasize the importance and novelty of their work, revise redundant descriptions in Results while focusing on demonstrating statistical significance. With careful revision, this manuscript has considerable potential to make a meaningful contribution to the field.

Response

Dear reviewer #1,

We sincerely thank you for your insightful and comprehensive comments, which have been very helpful in improving the quality and clarity of this manuscript. In response to your suggestions, we have undertaken substantial revisions.

Specifically, we revised the Introduction to more clearly articulate the scientific motivation, urgency, and novelty of forest-specific, high-resolution mapping of soil bulk density and pH across China. We also carefully refined the Results section to reduce redundancy and to emphasize quantitative interpretation supported by appropriate statistical evidence, avoiding subjective or over-interpreted statements.

We believe that these revisions substantially enhance the rigor, transparency, and interpretability of the manuscript. Our point-by-point responses to all comments are provided below, and all corresponding revisions are marked in **blue** in the revised manuscript.

Best regards,

Jizhen Chen

**Major comment 1**

First of all, after reading the Introduction, I wasn't fully convinced of the necessity and urgency of this study. The Introduction section begins with very basic background information on forest soil, which is too general to establish a compelling rationale. The excessive introduction about methodology doesn't effectively build a case for the study's significance, either. For instance, the entire second paragraph is basically saying "a lot of people have done this", which may justify methodological reliability but not why this work is needed. The fourth paragraph focuses on the historical development of methodologies, which isn't the main goal of an Introduction. While building a nationwide forest soil profile database is potentially valuable, the current Introduction does not sufficiently highlight how this study advances beyond simply extracting forest-covered data from existing maps.

**Response**

We thank the reviewer for this constructive and important comment. We agree that the original Introduction did not sufficiently establish the necessity and urgency of this study.

In response, we have substantially revised the Introduction to adopt a more problem-driven structure. General background information on forest soils and the historical development of digital soil mapping methodologies has been condensed (***Lines 37-60***). The revised text now explicitly emphasizes the limitations of existing national and global soil bulk density and pH products, which are largely derived from mixed-ecosystem samples and therefore fail to capture the distinct spatial heterogeneity and vertical structure of forest soils (***Lines 61-73***).

Importantly, we now clarify that simply extracting forest-covered pixels from existing soil maps is insufficient. Instead, we highlight the need for forest-specific modeling frameworks that explicitly account for ecosystem-specific processes and depth-dependent variability. This rationale is clearly articulated in ***Lines 79-80***, where we emphasize the ecological importance, spatial complexity, and current lack of high-resolution forest soil BD and pH estimates across China.

**Major comment 2**

Some findings are presented without statistical validation and therefore unconvincing. For example, L255 "BD prediction accuracy...peaking at intermediate depths (15–30 cm: MEC = 0.657) with lower accuracy in surface layers (0–5cm: MEC = 0.598) and deep layers (60–100 cm: MEC = 0.656)". Without testing for statistical significance, how can 0.656 represent "lower accuracy" compared to 0.657? Similarly, statements such as "all predictions maintained negligible bias (|ME| ≤ 0.019) across depth intervals" lack a defined threshold for "negligible". Descriptions like "Conversely, pH predictions demonstrated superior accuracy: CV maintained strong performance across depths" appear subjective, without definition for "superior" or "strong".

**Response**

Thank you for for your important comment regarding the interpretation of model performance metrics. In response, we have revised the manuscript to avoid over-interpretation of small numerical differences in performance indicators and to remove subjective descriptors that were not supported by formal statistical testing.

Specifically, statements comparing prediction accuracy across soil depths (e.g., "higher" or "lower" accuracy) have been removed, as differences in MEC values such as 0.656 versus 0.657 are not statistically meaningful. Similarly, qualitative terms such as "superior," "strong," and "negligible" have been replaced with objective descriptions based on the reported ranges of MEC, RMSE, and ME values.

The revised text now focuses on presenting model performance in a descriptive and transparent manner, emphasizing the overall consistency between cross-validation and independent validation results, as well as the absence of systematic bias indicated by ME values close to zero.

These revisions can be found in ***Section 3.2 (Lines 275–288)***.

**Major comment 3**

Similarly, in the Result section, the authors keep emphasizing that their "patterns align with former maps", which further raises questions about the novelty and importance of this work.

**Response**

We thank the reviewer for this insightful comment regarding the interpretation of similarities between our results and existing soil datasets. In response, we have revised the manuscript to avoid overemphasizing pattern agreement with previous products in the Results section.

Specifically, statements such as "Macroscale patterns align with existing maps" have been
removed from the Results, as simple visual consistency alone does not sufficiently reflect the
novelty or contribution of this work. Instead, we have substantially revised *Section 4.1 (Lines 451–*
*511)* to provide a more detailed and quantitative comparison with existing datasets (CSDLv2,
ChinaSoilInfoGrids, and SoilGrids 2.0).

This revised discussion focuses on ecosystem-specific differences in both the vertical
distribution and magnitude of forest soil BD and pH, highlighting discrepancies that are not captured
by generalized soil products. In particular, we demonstrate that existing datasets fail to fully
represent the non-linear depth-dependent pattern of forest soil BD and systematically predict higher
BD values in deeper layers (60–100 cm), which may lead to overestimation of forest soil carbon
stocks. Similarly, our results indicate consistently lower forest soil pH values compared to existing
datasets, suggesting that ecosystem-specific acidification processes in forest soils are
underestimated in generalized products.

These revisions clarify that the contribution of this study lies not in reproducing existing spatial
patterns, but in providing forest-specific, high-resolution estimates that improve the representation
of soil properties and associated ecological processes.

**Major comment 4**
Many descriptions in the Results section are excessive or repetitive (e.g., L268–270, L274–279),
and some qualitative statements regarding spatial gradients (e.g., "BD values increase from the coast
inland" in L271) are unclear.

**Response**
We thank the reviewer for pointing out that parts of the Results section were overly descriptive
and repetitive, and that some qualitative statements regarding spatial gradients lacked clarity. In
response, we have substantially revised the Results section to improve conciseness and clarity, and
to shift from generalized qualitative descriptions toward a more quantitative and statistically
supported characterization of spatial patterns.

Specifically, repetitive regional descriptions have been removed, and ambiguous statements
such as "BD values increase from the coast inland" have been eliminated. Instead, we now quantify
spatial patterns using latitudinal and longitudinal gradients, together with regional statistical
summaries (boxplots), which provide a clearer and more objective representation of spatial
variability.

To support this revision, Figures 3 and 4 have been redesigned (now *Figures 4 and 5*) to
explicitly illustrate depth-specific latitudinal and longitudinal trends, as well as regional differences
in BD and pH. The revised Results section (*Section 3.3, Lines 290–345*) now presents the main
spatial features more concisely, while preserving the key information needed to interpret large-scale
patterns.

**Major comment 5**
Why is FRFS introduced in the Introduction section but QRF in the Method?

**Response**
We thank the reviewer for this comment regarding the structure and consistency of the methodological description. We agree that the initial presentation may have given the impression that FRFS and QRF were introduced at different conceptual levels.

In response, we have revised the Introduction to explicitly present FRFS and QRF as complementary components of a unified modeling framework. Specifically, FRFS is introduced as a feature selection strategy designed to reduce dimensionality and improve model parsimony and interpretability, while QRF is described as the core predictive algorithm used to model soil BD and pH and to quantify prediction uncertainty.

This dual-focused strategy is now clearly outlined in the Introduction *(Lines 46–60)*. The Methods section then follows this conceptual structure, first detailing the covariate selection procedure based on FRFS *(Section 2.2.1)*, and subsequently describing the implementation of the QRF model as the primary predictive tool *(Section 2.2.2)*. This revision ensures consistency between the Introduction and Methods and clarifies the distinct but integrated roles of FRFS and QRF within the overall DSM framework.

**Major comment 6**

Table 1 may be presented more clearly as a figure, and currently has a confusing caption.

**Response**

We thank the reviewer for this helpful suggestion regarding the presentation of Table 1. In response, we have revised the manuscript to improve the clarity and interpretability of this information. Specifically, Table 1 has been converted into a violin plot (now *Fig. 3*), which more effectively illustrates the distribution of soil samples across depth intervals and highlights differences between layers.

In addition, statistical tests have been applied to assess the significance of differences among soil depths, and the corresponding results are now explicitly shown in the figure. The original table has been moved to the Supplementary Information *(Table S5)* for reference. The Results section has been updated accordingly to reflect the revised figure and the additional statistical information *(Section 3.1, Lines 248–274)*. These changes improve the clarity of data presentation and provide a more informative summary of the sampling structure.

**Major comment 7**

Figure 6 might benefit from an overall analysis across depths, and consider adding relationships between BD and MAP (or other key covariates) in supplementary materials.

**Response**

We thank the reviewer for this constructive suggestion regarding the analysis of Figure 6 and the relationships between soil properties and key covariates.We agree that an overall comparison across soil depths, together with a clearer interpretation of the relationships between BD/pH and major environmental drivers, would substantially strengthen the manuscript. In response, we have extended the methodological framework by introducing SHAP (SHapley Additive exPlanations) analysis *(Section 2.3, Lines 222–229)*.

This addition addresses a key limitation of the relative variable importance measures previously derived from the QRF models, which reflect covariate importance only in a relative sense within each depth-specific model and depend on the selected feature set.

Because FRFS yields different covariate subsets for different soil layers, these relative importance values are not directly comparable across depths. SHAP provides independent, additive contribution scores for each predictor, enabling consistent cross-depth comparison and allowing both the magnitude and direction of covariate effects to be quantitatively interpreted. Based on this approach, Figures 6 and 7 have been revised and are now presented as *Figures 7 and 8*, illustrating depth-consistent importance patterns and the relationships between BD/pH and key covariates.

Accordingly, the previous Results subsection on variable importance has been fully revised and replaced by *Section 3.5 (Lines 377–453)*, which now presents the SHAP-based analysis and interpretation in place of the original QRF relative importance results.

**Major comment 8**

L85 & 91, QRF should be explained upon its first mention.

**Response**

We thank the reviewer for pointing out this issue. In response, we have revised the manuscript to ensure that Quantile Regression Forest (QRF) is fully explained at its first mention *(Lines 51)*. In addition, we conducted a systematic check of abbreviations throughout the manuscript to ensure consistent definition and usage upon first appearance.

**Major comment 9**

Abbreviations (including BD, SD and the abbreviations of models) in Tables and Figures should be clearly defined in their captions to make them self-explanatory.

**Response**

We thank the reviewer for this helpful comment. In response, we have revised all figure and table captions to ensure that abbreviations (including BD, SD, and model abbreviations) are clearly defined upon first appearance, making the tables and figures self-explanatory. This change has been applied consistently throughout the manuscript.

**Major comment 10**

L111 is redundant with L108.

**Response**

We thank the reviewer for carefully identifying this redundancy. In response, we have revised the manuscript to improve clarity and conciseness by removing the sentence.

**Major comment 11**

L251, rephrase "conversely".

**Response**

We thank the reviewer for this helpful suggestion regarding wording. In response, we have revised the relevant sentence and removed the use of "conversely," which was no longer appropriate given the revised structure and interpretation of the Results section. Following the substantial revision of ***Section 3.2***, the description of pH model performance has been rewritten to avoid subjective or contrastive wording and to present the results in a more neutral and consistent manner based on the reported performance metrics.

---

## Author Comment (AC2)

**RESPONSE TO REVIEWER #2**

The authors address a significant topic by mapping key soil properties across China's forests using a comprehensive dataset. The resulting high-resolution products have the potential to be a valuable resource for the scientific community. However, several methodological and descriptive aspects require substantial improvement to ensure the reliability and reproducibility of the findings.

Response
Dear reviewer #2,
We sincerely thank you for your insightful and comprehensive comments, which have been very helpful in improving the rigor, transparency, and reproducibility of this manuscript. Following your suggestions, we have undertaken substantial revisions.

Specifically, we strengthened the methodological description and expanded the Discussion to more explicitly address sources of uncertainty (e.g., covariate resolution harmonization) and model interpretability across soil depths. We also added new supplementary materials to improve transparency, including covariate maps, sample distribution summaries, and additional quantitative comparisons with existing BD and pH products.

Our point-by-point responses are provided below. All corresponding revisions are marked in **blue** in the revised manuscript.

Best regards,
Jizhen Chen

**General comments:**

**General comments 1**

Resampling covariates with diverse native resolutions to a 90-m grid introduces significant uncertainty, particularly for inputs derived from coarser scales. This issue warrants a detailed discussion and quantification to assess the reliability of the final high-resolution maps.

**Response**

We thank the reviewer for raising this important point regarding uncertainty introduced by harmonizing environmental covariates with heterogeneous native resolutions to a 90-m grid.

In this study, all covariate layers were projected to a common coordinate system and resampled to 90 m using bilinear interpolation, following common practice in national-scale digital soil mapping and consistent with established products (Poggio et al., 2021; Liu et al., 2022; Shi et al., 2025). We acknowledge that resampling, particularly from coarser-resolution inputs, may introduce scale-related uncertainty and smoothing effects, and that the effective spatial resolution of the final predictions is constrained by the coarsest covariates.

In response, we have expanded the Discussion to explicitly address this limitation and its implications for map interpretation *(Section 4.3, Lines 538–551)*. As noted there, isolating and quantifying uncertainty attributable solely to covariate resampling is methodologically challenging at national scales and is rarely reported in existing large-area DSM studies. Therefore, the resulting maps should be interpreted as conditional estimates that represent the most probable spatial patterns given the available covariates and their effective spatial support, rather than as direct representations of fine-scale soil heterogeneity at 90 m.

Reference:

Liu, F., Wu, H., Zhao, Y., Li, D., Yang, J., Song, X., Shi, Z., Zhu, A., and Zhang, G.: Mapping high
resolution national soil information grids of China, Sci. Bull., 67, 328–340,
https://doi.org/10.1016/j.scib.2021.10.013, 2022.

Poggio, L., De Sousa, L. M., Batjes, N. H., Heuvelink, G. B. M., Kempen, B., Ribeiro, E., and
Rossiter, D.: SoilGrids 2.0: producing soil information for the globe with quantified spatial
uncertainty, Soil, 7, 217–240, https://doi.org/10.5194/soil-7-217-2021, 2021.

Shi, G., Sun, W., Shangguan, W., Wei, Z., Yuan, H., Li, L., Sun, X., Zhang, Y., Liang, H., Li, D.,
Huang, F., Li, Q., and Dai, Y.: A China dataset of soil properties for land surface modelling
(version 2, CSDLv2), Earth Syst. Sci. Data, 17, 517–543, https://doi.org/10.5194/essd-17-517-
2025, 2025.

**General comments 2**

The manuscript provides insufficient discussion on the variable importance for BD and pH across
different soil layers. The underlying reasons for these variations require further elaboration.

**Response**

We thank the reviewer for this comment. We agree that the original manuscript did not sufficiently
elaborate on depth-dependent variations in variable importance and the underlying reasons for these
patterns.

In response, we substantially revised the interpretability analysis by introducing SHAP (SHapley
Additive exPlanations), which enables consistent, depth-comparable quantification of both the
magnitude and direction of covariate effects. This revision replaces the previous discussion based
on QRF-derived relative variable importance, which primarily supports within-model comparisons
and is not directly comparable across depths when the selected feature sets differ.

The revised *Section 3.5* now provides a depth-explicit interpretation of key drivers for both BD and
pH, and the corresponding methodological update is described in *Section 2.3*. The updated results
are presented in *Figures 7 and 8*.

**General comments 3**

Natural and planted forests possess distinct driving mechanisms. Developing separate models for
each forest type is advisable to accurately capture these specific variations.

**Response**

We thank the reviewer for this insightful suggestion. We agree that natural and planted forests often
differ in management history, stand structure, and species composition, which may influence soil
processes and their responses to environmental drivers.

However, in our compiled forest soil profile database, forest origin (natural vs. planted) is not
consistently documented in the original metadata, which prevents a reliable partition of the full
dataset by forest type. To evaluate the potential impact of forest origin, we conducted an additional
targeted test using an external forest-type dataset to assign forest origin at sampling locations (Cheng
et al., 2024), and performed separate modeling for the 60–100 cm layer as a representative case.

The results showed a pronounced decline in predictive performance when models were
separated by forest origin, primarily due to substantially reduced sample sizes at this depth (e.g.,
BD: planted MEC = 0.462, RMSE = 0.598; natural MEC = 0.302, RMSE = 0.632; pH: planted MEC
= 0.503, RMSE = 0.482; natural MEC = 0.404, RMSE = 0.402).
Moreover, evidence from the literature suggests that, at regional to national scales, the
dominant controls on soil bulk density and pH are largely consistent across natural and planted
forests. Broad environmental gradients—particularly climate, parent material, and long-term soil
development processes—have been shown to exert primary control on soil physical and chemical
properties, while forest origin mainly modulates response magnitude through differences in stand
structure or management intensity rather than altering the fundamental driver mechanisms
(Luyssaert et al., 2008; Pretzsch et al., 2014) .
Accordingly, separating models by forest origin at the national scale is unlikely to substantially
improve predictive performance for BD and pH, while potentially increasing uncertainty due to
reduced sample sizes. Given these considerations, together with the national-scale objective of this
study, we retained a unified modeling framework to ensure robust, spatially consistent predictions
across China's forest domain.
Reference:

Cheng, K., Yang, H., Guan, H., Ren, Y., Chen, Y., Chen, M., Yang, Z., Lin, D., Liu, W., Xu, J., Xu,
G., Ma, K., and Guo, Q.: Unveiling China's natural and planted forest spatial-temporal
dynamics from 1990 to 2020, ISPRS J. Photogramm. Remote Sens., 209, 37–50,
https://doi.org/10.1016/j.isprsjprs.2023.11.003, 2024.
Luyssaert, S., Schulze, E. D., Börner, A., Knohl, A., Hessenmöller, D., Law, B. E., Ciais, P., and
Grace, J.: Old-growth forests as global carbon sinks, Nature, 455, 213–215,
https://doi.org/10.1038/nature07276, 2008.
Pretzsch, H., Biber, P., Schütze, G., Uhl, E., and Rötzer, T.: Forest stand growth dynamics in Central
Europe have accelerated since 1870, Nat. Commun., 5, 4967,
https://doi.org/10.1038/ncomms5967, 2014.

**General comments 4**
Providing spatial distribution maps for every covariate listed in Table S1 is advisable. The figures
need to clearly display the value ranges for continuous variables and the distinct spatial patterns for
each category within categorical variables. Specifying the number of sample points in both the
training and validation sets for each categorical variable is recommended.

**Response**
We thank the reviewer for this constructive suggestion regarding the presentation of environmental
covariates and sample stratification.
In response, we have added spatial distribution maps for all environmental covariates retained
after FRFS selection, which are now provided in the ***Supplementary Information (Fig. S1)***. These
figures display the value ranges of continuous variables and the spatial patterns of categorical
variables, thereby improving transparency and interpretability of the covariate inputs.
In addition, we have quantified the number of soil sampling plots associated with each category
of the categorical covariates separately for the training and independent validation datasets, using the surface soil layer (0–5 cm) as a representative overview. These statistics are summarized in *Tables S2–S4 in the Supplementary Information.*

Together, these additions enhance the reproducibility of the study and provide a clearer basis for assessing the robustness of the modeling framework.

**General comments 5**

Forest age represents a critical covariate. Incorporating this variable into the analysis is advisable to improve model performance.

**Response**

We thank the reviewer for this important suggestion regarding the inclusion of forest age as a covariate. We fully agree that forest age is a critical variable influencing forest structure, productivity, and soil processes, and therefore has the potential to improve model performance when reliable and spatially consistent data are available.

During the initial covariate selection stage, we carefully evaluated commonly used forest age datasets for China (Cheng et al., 2024). However, currently available forest age maps are derived products and still exhibit substantial uncertainty at local scales, particularly in cold-temperate coniferous forests, where reported model performance remains relatively low ($R^2 = 0.47$).

More importantly, existing forest age products are developed based on forest domain definitions and source data that differ from those adopted in this study. As a result, their spatial coverage is not fully consistent with the forest domain considered here, leading to notable spatial mismatches and discontinuities when the two datasets are overlaid. Integrating forest age products into a national, wall-to-wall digital soil mapping framework would therefore introduce systematic spatial bias and compromise prediction consistency. Given these considerations, we did not include forest age as a covariate in the present study.

We note that digital soil mapping under conditions of incomplete or spatially inconsistent covariate coverage represents a methodological challenge rather than a purely data-related issue. This challenge is particularly pronounced in national-scale forest soil mapping studies in China, where heterogeneity in data sources and definitions is common. Recent methodological advances, such as the iPSM-based framework proposed by Fan et al. (2020), have demonstrated that covariate incompleteness can be addressed by explicitly accounting for missing variables and their associated uncertainty. Accordingly, we acknowledge this limitation in the Discussion (Show in *Section 4.3. Line 553-568*) and consider the further development and application of such approaches to be an important direction for improving future forest soil mapping frameworks.

Reference:

Cheng, K., Yang, H., Guan, H., Ren, Y., Chen, Y., Chen, M., Yang, Z., Lin, D., Liu, W., Xu, J., Xu, G., Ma, K., and Guo, Q.: Unveiling china's natural and planted forest spatial–temporal dynamics from 1990 to 2020, ISPRS J. Photogramm. Remote Sens., 209, 37–50, https://doi.org/10.1016/j.isprsjprs.2024.01.024, 2024.

Fan, N. Q., Zhu, A.-X., Qin, C.-Z., and Liang, P.: Digital soil mapping over large areas with invalid environmental covariate data, ISPRS Int. J. Geo-Inf., 9, 102, https://doi.or g/10.3390/ijgi9020102, 2020.

**General comments 6**

Providing the original data is necessary to facilitate the reproducibility of the study by other researchers.

**Response**

We thank the reviewer for emphasizing the importance of reproducibility. We would like to clarify that the original forest soil profile data used in this study are subject to data confidentiality agreements and access restrictions imposed by the institutions responsible for the national forest soil survey in China, and therefore cannot be publicly released.

Nevertheless, we have made all data that can be legally shared openly available, and access to restricted data may be requested by contacting the corresponding author, subject to the relevant data-sharing policies. To ensure the highest possible level of reproducibility under these constraints, we provide comprehensive documentation of the data processing workflow, quality control and harmonization procedures, environmental covariates, modeling framework, and validation strategy.

In addition, all derived products generated in this study, including gridded maps of soil bulk density and pH and their associated uncertainty estimates, are openly available through a public repository.

We believe that this level of transparency allows other researchers to fully reproduce the methodology and apply it to independent datasets, while respecting the access restrictions governing the original observations.

**General comments 7**

Comparing the current results with existing soil BD and pH products is recommended. The manuscript needs to clarify specific improvements and explain the reasons for these advancements.

**Response**

We thank the reviewer for this suggestion. Following it, we conducted additional quantitative comparisons between our forest-specific predictions and existing national and global datasets, including CSDLv2, ChinaSoilInfoGrids, and SoilGrids 2.0. These analyses explicitly quantify differences in predicted BD and pH across depths and regions, and clarify where and why forest-specific mapping provides added value.

The statistical comparisons are summarized in *Figures S5 and S6 at Supplementray Information*, and spatial contrasts are shown in *Figures 9 and 10*. The results indicate that our forest-specific maps better capture depth-dependent patterns and ecosystem-specific magnitude differences that are not well represented by generalized products. The underlying reasons for these improvements are discussed in *Section 4.1*, including the use of forest-only observations, forest-specific covariate–response relationships, and depth-resolved modeling.

**General comments 8**

Presenting the spatial distribution of sample points for both the training and validation sets is necessary. The manuscript should also address whether these distributions are spatially balanced.

**Response**

We thank the reviewer for this suggestion regarding the spatial distribution and balance of the training and validation samples.

In response, we have provided a new figure *(Fig. S2)* that visualizes the spatial distributions of the training and independent validation datasets for BD and pH across all soil layers, supporting the spatial balance claim.

Additionally, we have added a statement in the Methods section *(Lines 97–98)* to clarify that the spatial distributions of the training and validation subsets were explicitly examined to ensure that both subsets maintain spatial balance across regions and soil depths.

**General comments 9**

A more detailed description of the raw data is necessary. The manuscript should specify the sample sizes and spatial distributions across different temporal periods, soil types, and forest types.

**Response**

We thank the reviewer for requesting a more detailed description of the raw data. In response, we revised the data description to explicitly report sample sizes and their distributions across forest types, soil types, and temporal periods. Detailed counts by forest type, soil type, and related categories are provided in *Tables S2–S5* (referenced in the main text; *Lines 112*).

In addition, we clarified the temporal coverage of the dataset by reporting the number of soil profiles collected in each survey year from 2018 to 2023 (*Lines 119–120*). These revisions provide a more transparent overview of the dataset composition in terms of time, forest types, soil types, and spatial coverage.

**Minor comments:**
**Minor comments 1**

Lines 1-3: The general phrase "Soil properties" creates redundancy with the specific variables "Bulk Density and pH," necessitating a more concise revision such as "High-Resolution, Multi-Depth Mapping of Soil Bulk Density and pH in China's Forests Using Machine Learning".

**Response**

We thank the reviewer for this helpful suggestion. We revised the title by removing the general phrase "Soil properties" and focusing on the specific variables examined in this study. The revised title is: "High-Resolution, Multi-Depth Mapping of Soil Bulk Density and pH in China's Forests Using Machine Learning." This change is reflected in *Lines 1–3.*

**Minor comments 2**

Lines 20-21: The claim of being 'first' is inaccurate due to the existence of prior 90-m products, so the text should be revised to focus on the specific contribution to forest ecosystems instead. and Lines 429-430: The use of "first" is an absolute claim that is prone to dispute.

**Response**

We thank the reviewer for pointing out that the term "first" is an absolute claim and may be disputable given the existence of prior 90-m soil products. We agree and have revised the text to avoid absolute wording, shifting the emphasis to the specific contribution of this study—namely, forest-specific mapping and depth-resolved estimates of BD and pH. The relevant revisions have been made in ***Lines 18*** and ***Lines 598–599***.

**Minor comments 3**

Lines 73-74: The phrase "in heterogeneous" is grammatically incorrect.

**Response**

We thank the reviewer for identifying this grammatical issue. The incorrect phrase has been removed during the revision of the Introduction, and the corresponding sentence has been rewritten for clarity.

**Minor comments 4**

Lines 109-111: This sentence is redundant and should be deleted.

**Response**

We thank the reviewer for pointing out this redundancy. We deleted the redundant sentence as suggested and revised the surrounding text accordingly.

**Minor comments 5**

Lines 111-112: Specify the quality control and data harmonization methods.

**Response**

We thank the reviewer for this suggestion. We have added a clearer description of the quality control and data harmonization procedures (***Lines 115–116***), including unit standardization, harmonization of depth intervals, and reconciliation of metadata across survey sources.

**Minor comments 6**

Lines 139-155: The 41 environmental covariates lack necessary citations, and the sources or the data itself should be made accessible to readers to ensure reproducibility.

**Response**

We thank the reviewer for this comment. In response, we added detailed citations and source information for all 41 environmental covariates in ***Supplementary Table S1*** to improve transparency and reproducibility.

**Minor comments 7**

Lines 159-160: The number of standardized soil layers should be corrected from four to five.

**Response**

We thank the reviewer for pointing out this discrepancy. We corrected the number of standardized soil layers from four to five ***(Line 141)***.

**Minor comments 8**

Lines 222-223: $q50$ denotes the median prediction.

**Response**

We thank the reviewer for this correction. We revised the manuscript to clarify that q0.50 denotes the median prediction and corrected the notation accordingly (***Line 185***). We confirm that this was a notation issue only; all analyses were conducted using the median (q0.50) prediction, and the correction does not affect the results or conclusions.